# Development and Characterization of Nano-Ink from Silicon Carbide/Multi-Walled Carbon Nanotubes/Synthesized Silver Nanoparticles for Non-Enzymatic Paraoxon Residuals Detection

**DOI:** 10.3390/mi14081613

**Published:** 2023-08-16

**Authors:** Itsarapong Chuasontia, Wichaya Sirisom, Natthapon Nakpathomkun, Surachet Toommee, Chiravoot Pechyen, Benchamaporn Tangnorawich, Yardnapar Parcharoen

**Affiliations:** 1Department of Physics, Faculty of Science and Technology, Thammasat University, Bangkok 12120, Thailand; itsarapong.ch@lsed.tu.ac.th (I.C.);; 2Faculty of Learning Science and Education, Thammasat University, Bangkok 12120, Thailand; 3Thammasat University Center of Excellence in Modern Technology and Advanced Manufacturing for Medical Innovation, Thammasat University, Bangkok 12120, Thailand; 4Department of Materials and Textile Technology, Faculty of Science and Technology, Thammasat University, Bangkok 12120, Thailand; 5Industrial Arts Program, Faculty of Industrial Technology, Kamphaeng Phet Rajabhat University, Kamphaeng Phet 62000, Thailand; 6Chulabhorn International College of Medicine, Thammasat University, Bangkok 12120, Thailand

**Keywords:** silicon carbide, silver nanoparticle, non-enzymatic electrochemical sensor, pesticide sensor, paraoxon

## Abstract

The ongoing advancement in the synthesis of new nanomaterials has accelerated the rapid development of non-enzymatic pesticide sensors based on electrochemical platforms. This study aims to develop and characterize Nano-ink for applying organophosphorus pesticides using paraoxon residue detection. Multi-walled carbon nanotubes, silicon carbide, and silver nanoparticles were used to create Nano-ink using a green synthesis process in 1:1:0, 1:1:0.5, and 1:1:1 ratios, respectively. These composites were combined with chitosan of varying molecular weights, which served as a stabilizing glue to keep the Nano-ink employed in a functioning electrode stable. By using X-ray powder diffraction, Raman spectroscopy, energy dispersive X-ray spectroscopy, and a field emission scanning electron microscope, researchers were able to examine the crystallinity, element composition, and surface morphology of Nano-ink. The performance of the proposed imprinted working electrode Nano-ink was investigated using cyclic voltammetry and differential pulse voltammetry techniques. The Cyclic voltammogram of Ag NPs/chitosan (medium, 50 mg) illustrated high current responses and favorable conditions of the Nano-ink modified electrode. Under the optimized conditions, the reduction currents of paraoxon using the DPV techniques demonstrated a linear reaction ranging between 0.001 and 1.0 µg/mL (R^2^ = 0.9959) with a limit of detection of 0.0038 µg/mL and a limit of quantitation of 0.011 µg/mL. It was concluded that the fabricated Nano-ink showed good electrochemical activity for non-enzymatic paraoxon sensing.

## 1. Introduction

In Thailand, the agricultural industry is an essential economic driver and exporter of industrially processed food and agricultural products such as rice, rubber, sugar, tropical fruit, and cassava, which are demanded in the country and exported overseas. To increase productivity, product quality, and product appearance in response to market demand, reduce crop production losses from numerous pests and disease carriers such as mosquitoes, ticks, rodents, and mice, and increase crop production levels, weeds, insect infestations, and illnesses are all controlled through the use of pesticides in agriculture. Each type of pesticide, such as organophosphates, carbamates, organochlorine insecticides, and pyrethroids, is designed to be efficient against a particular pest. Organophosphates, carbamates, and herbicides accounted for most pesticides found in patients in Thailand, but 85–90% of all cases could not be linked to a specific pesticide [1,2,3].

Organophosphorus pesticides (OPs) worldwide are intensively used for pest control in many crops, including cereals, fruits, coffee, potatoes, and sugarcane [4,5]. Due to their low solubility and environmental degradation, pesticides are also known as persistent organic pollutants. As a result, they are intrinsically harmful because their buildup in living things can result in catastrophic illnesses [6]. The OP’s poison obstructs the peripheral nervous system, affecting various muscles, glands, and smooth muscles. Symptoms include dizziness, seizures, muscular hypotonia, and muscle cramps [7]. Moreover, multiple studies have shown that human exposure to OPs can aberrate embryonic development and cause neurodegenerative diseases such as Alzheimer’s and motor neuron disease [8]. However, pesticides are well known for their toxicity. Some pesticides have low toxicity, and it would take significant exposure to harm the user. Other pesticides have high toxicity, and a little exposure could seriously damage the user immediately or over the long term [9,10]. Nowadays, various analytical techniques, such as gas chromatography coupled with mass spectrometry (GC-MS, GC-MS/MS) and electron impact ionization (EI) or liquid chromatography-tandem mass spectrometry (LC-MS/MS) with electrospray ionization (ESI), are commonly used for multi-residue pesticide detection in food due to their excellent sensitivity and selectivity, as well as their ability to screen multiple pesticides from various chemical classes in very complex matrixes in a single run [11]. Although these methods promise accurate and sensitive detection of OPs, they require complicated preparation processes and expensive equipment and are slow to detect and operate in the laboratory. Consequently, electrochemical techniques have been considered for detecting pesticides because they provide rapid responses, simple processes, and high sensitivity [12,13].

Furthermore, the amount of pesticide is very low in the target sample, so the sensitivity of a detection method is limited to detecting pesticides directly using electrochemical techniques. Thus, detection methods need to be modified. Research has described how to improve the sensitivity and efficiency of detection methods. For example, a working electrode was enhanced with a composite film for sensing methyl parathion based on a nano-titanium and graphene composite film-modified glassy carbon electrode [14] or nanomaterial ink for on-site direct detection of parathion residues from vegetable products, which does not require high temperature or vacuum filtration [15]. Therefore, electrochemical materials are necessary for electrode fabrication to improve the sensitivity of pesticide detection.

Silicon carbide (SiC)-modified electrodes receive more attention in electrochemical analysis due to their stable physical and chemical properties, high electrical conductivity, high aspect ratio, low background current, excellent catalytic oxidation, good biocompatibility, good absorption properties, marked electrocatalyst activity, and lack of toxicity [13]. In recent work, it has been reported that the construction of an electrochemical sensor made from a silicon carbide-modified glassy carbon electrode for the direct detection of parathion shows a high response rate, a significant wide linear detection range, and a solid ability to adsorb parathion pesticides [16,17]. In addition, SiC was used to modify the working electrode to detect quinalphos (QNP) by electroanalytical techniques. Due to the excellent properties of SiC, such as its high aspect ratio, good conductivity, and high electrocatalyst activity, the modified electrode had excellent sensitivity, a wide linear detection range, and a good nanomolar detection limit [18].

Drop casting was indicated as the most convenient technique for nanomaterial deposition on the electrode surface in terms of usefulness on the laboratory scale, considering the binders that immobilized suspension on the electrode surface. Chitosan was often used as a binder for electrode preparation with the drop casting method [19,20,21]. Chitosan biopolymer is a cationic polyelectrolyte and is hydrophilic. The chitosan property is an appropriate polymer for mitigating the coffee-ring effect that can be achieved, which can be beneficial for ensuring diffusion linearity and electrode reproducibility [20]. The various properties of chitosan include nontoxicity, good adhesion with excellent film-forming ability, high mechanical strength, high permeability, and excellent adsorption capability of metal ions, which are favorable for sensing platforms [22,23,24]. Moreover, differences in molecular weights and degrees of deacetylation affect physical and chemical properties [25]. For instance, in this report, they investigated the impact of chitosan nanoparticles’ molecular weight and their use in drug administration. The drug loading was highest when the chitosan DD was 90% and its total Mw value, composed of three different molecular weights of chitosan, was 55 kDa. Because the shorter chitosan fragments in the chitosan solution had a lower viscosity and average molecular weight, their free amino groups were simpler to protonate, which made ionic interactions more effective at adsorbing 5-Fluorouracil (5-FU) [26]. However, the chitosan solution also has poor thermal stability and non-conducting characteristics. Adding some nanoparticles can improve chitosan bioadhesion due to metal oxide’s unique ability to promote fast electron transfer kinetics between the electrode and the electrolyte [27,28].

Composite carbon-based nanomaterials, such as multi-walled carbon nanotubes (MWCNTs), were used to fabricate flexible conducting electrodes for sensor applications due to their unique porous network, higher surface area, excellent electron transport rates, and robust mechanical properties. However, a metallic nanoparticle composite was used to facilitate the transfer of electrons between the solution and the surface of the electrodes, leading to an enhanced signal and improved sensor sensitivity [29,30].

Furthermore, silver nanoparticles (AgNPs) have been widely used in electrode modification because of their unique properties, such as high conductivity, faster electron communication features, and good chemical and bio-sensing abilities [21,31]. Silver nanoparticles can be synthesized using different physical and chemical methods. The high production costs and toxicity problems of earlier approaches are what led to the development of a dependable, cost-efficient, and environmentally friendly green synthesis technique [32]. In this technique, the AgNPs are prepared using corn starch as a reducing and stabilizing agent [33].

In this study, the fabrication of inks from multi-walled carbon nanotubes, silicon carbide, silver nanoparticles, and chitosan, which has two molecular weights (medium: 125–500 kDa and high: 500–900 kDa), was carried out and characterized by FE-SEM/EDS, XRD, Raman spectroscopy analysis, and electrochemical measurement, which was investigated by CV and DPV to determine the sensitivity of the fabrication ink against organophosphorus.

## 2. Materials and Methods

### 2.1. Reagents and Chemicals

Multi-walled carbon nanotubes (3–12 µm, 99.99%) were purchased from NANOGENSTORE (Chiang Mai, Thailand), and silicon carbide (50 nm, 99.9%) from Sigma-Aldrich (St. Louis, MO, USA). Silver nitrate (AgNO_3_, 99.9%) and Cetyltrimethylammonium bromide (CTAB, 99%) were purchased from Virotevittayapun Company Limited (Nakhon Ratchasima, Thailand). Corn starch was purchased at any market. Potassium hexacyanoferrate (II): K_4_(Fe(CN)_6_)·3H_2_O: Fe(II) was purchased from Sigma Aldrich. Potassium chloride (max. 0.0001%A) was purchased from Riedel de Haen. Chitosan (Mw = 125–500 kDa and 500–900 kDa, %DD = 90–95%) was purchased from BIO21 (Thailand). Acetic acid (99.7%) was purchased from APEX CHEMICALS (Singapore).

### 2.2. Apparatus and Method

The surface morphology and agglomeration were characterized using a field emission scanning electron microscope (FE-SEM; Model JEOL JSM7800 F, Akishima, Japan) equipped with energy dispersive X-ray spectroscopy (EDS; Model Oxford XMax 20, Oxford, UK). The crystalline structure was confirmed by the X-ray Diffractometer (XRD; Bruker AXS Model D8 Advance, Berlin, Germany). Raman measurements were carried out with a Thermo Fisher Scientific DXR3 Raman Microscope (BEC Co., Ltd., Bangkok, Thailand) at 532 nm wavelength and 900 lines/mm of grating. The electrochemical measurements of cyclic voltammetry (CV) and differential pulse voltammetry (DPV) were performed on Plamsense BV (Chulabhorn International College of Medicine, Thammasat University, Bangkok, Thailand) equipped with PSTrace software (v. 5.9) interfaced to a personal computer. A conventional three-electrode system was used, consisting of a screen-printed carbon electrode with a diameter of 12.5 × 30 mm. A system was employed consisting of a working electrode made of silicon nanoparticles and multi-walled carbon nanotube ink, carbon as the counter electrode, and an Ag/AgCl electrode saturated with KCl as the reference electrode. CVs were recorded between −0.3 and 0.8 V with a scan rate of 10 mV s^−1^. DPV scans were recorded from 0.4 to −0.4 with a scan rate of 25 mV s^−1^, a pulse period of 0.2 s, and an E step of 0.01 V.

### 2.3. Preparation of Corn Starch Solution

Corn starch powder was prepared by dissolving a gram of powder in 10 mL of distilled water and stirring for 1 h at 100 °C. These starch solutions were used for the synthesis of silver nanoparticles.

### 2.4. Synthesis of Silver Nanoparticles

The synthesis of silver nanoparticles in a flower-shaped form was prepared by dissolving 0.33 mL of silver nitrate (AgNO_3_) in deionized water and mixing it with a cetyltrimethylammonium bromide (CTAB) solution, which was prepared by dissolving 10 mL of 0.1 M CTAB in deionized water. Then, 0.46 mL of corn starch solution was added, mixed, stirred, and kept at 40 °C in a water bath for 1 h.

For growth solution, 3 mL of silver nitrate (AgNO_3_) (15 mM) was dissolved in deionized water and mixed with cetyltrimethylammonium bromide solution, which was prepared from 2.5 mL of cetyltrimethylammonium bromide (CTAB) (0.2 M) dissolved in 10 mL of deionized water. Then 8 mL of cornstarch solution was added and mixed. Then 10 mL of solution from the first step was added to silver nitrate, mixed by stirring, and kept at 40 °C in a water bath for 24 h. After 24 h, the silver nitrate solution was centrifuged at 15,000 rpm for 30 s, poured on a petri dish, dried in an incubator at 90 °C for 20 min, and ground into a fine powder by mortar and pestle and ball mill for 5 h. The silver nanoparticle powder was placed in a vacuum oven at 90 °C.

### 2.5. Preparation of Nano-Ink for the Working Electrode

The ink for the working electrode was prepared by dispersing 50 mg of multi-walled carbon nanotubes, silicon carbide, and silver nanoparticles in different ratios in 10 mL of chitosan solution, as shown in Table 1. The chitosan solution was prepared by mixing 50 mg of high molecular weight (500–900 kDa) or medium molecular weight (125–500 kDa) chitosan in 10 mL of 0.1 M acetic acid, then ultrasonicating it for 30 min at 60 °C.

### 2.6. Preparation of Working Electrode and Pesticide on a Screen-Printed Carbon Electrode

A commercial screen-printed carbon electrode consists of three electrodes: the working and counter electrodes, which are carbon bases, and the reference electrode, which is Ag/AgCl. Improved working electrodes were prepared by drop casting ink on the surface of the commercial working electrode. Drop 2 µL of ink on the center of the working electrode, then dry it in an oven at 70 °C for 10 min. Figure 1A (right) shows a microscope image (Leica DM750 M (Wetzlar, Germany)) (objective lens magnification, ×10) of deposited ink after drying on a commercial electrode surface. The tiny particles on the surface are composite particle clusters of SiC. In Figure 1B, paraoxon solution prepared in different concentrations was dropped on a screen-printed carbon electrode to detect pesticides by DPV.

## 3. Results and Discussion

### 3.1. Structural and Morphological Studies

#### 3.1.1. X-ray Diffraction (XRD)

The crystalline nature of fabrication inks was confirmed by the X-ray Diffractometer with a pattern in the 2θ range from 20° to 80°. The dried powder of fabrication inks and raw materials prepared the sample for XRD measurement. Figure 2A lines (a–b) show XRD spectra of (a) MWCNTs/SiC/AgNPs 1:1:0-Cs (High) and (b) MWCNTs/SiC/AgNPs 1:1:0-Cs (Med). Sharp diffraction peaks in the planes (111), (220), and (002) of SiC correspond to Figure 2A line (g), MWCNTs, and Figure 2B line (h), respectively. In addition, the XRD spectra of MWCNTs/SiC/AgNPs with the ratios of 1:1:1-Cs (High), 1:1:1-Cs (Med), and 1:1:0.5-Cs (High) showed peaks at 2θ = 32.14°, 44.42°, and 64.55°, respectively, corresponding to the planes (111), (200), and (220), respectively, of the silver nanoparticle face-centered cubic structure. Moreover, the diffraction peaks appeared at 2θ = 25.760, corresponding to the plane (002), showing the crystalline graphite structure of MWCNT in the range of 2θ from 20° to 30°. Figure 2A lines (a–e) show disappearing peaks of chitosan because of the crystallite structure of SiC and AgNPs, whose XRD spectra are much sharper and highly intensified compared to chitosan. The pure chitosan peak is shown in Figure 2B lines (i–j) [34].

#### 3.1.2. Raman Spectroscopy

Raman analysis was used to study the presence of defects and crystal structure in the case of CNTs. Figure 3 shows the Raman spectra of (a) pure MWCNTs, (b) pure SiC, (c) pure AgNPs, (d) 1:1:0-Cs (High), and (e) 1:1:1-Cs (High). The stretching of the C-C bond in graphitic materials gives rise to the so-called G-band Raman feature, common to all sp2 carbon systems. The G-band indicates stress caused by external forces acting on the structure of multi-walled carbon nanotubes. G-bands are observed at approximately 1580 cm^−1^. Another important band in the Raman spectra of the investigated nanotubes at around 1350 cm^−1^ was observed, known as the D-band. The disordered structure of graphene sheets causes the D-band, and the 2D-band is approximately 2700 cm^−1^ [35,36], in which only the D- and G-bands of carbon were detected. The Raman shift spectra of SiC are observed at 790, 970, and 1500 cm^−1^. The fabricated ink showed SiC peaks disappearing; this is because the Raman scattering of carbon is at least 10 times higher than that of SiC [37,38]. After adding AgNPs, it was observed that its intensity is further enhanced in the case of AgNps/MWCNT composite, assuring the attachment of AgNPs as more defects have been incorporated by the binding of AgNPs onto MWCNT, justifying the synthesis of AgNPs/MWCNT/Cs composite.

The Raman spectra peak of AgNPs is 1292 and 1544 cm^−1^, which are obtained due to carboxylic symmetric and anti-symmetric C=O stretching vibrations of the carboxylic group that are formed by the oxidation of primary hydroxyl groups due to the interaction with silver NPs [39,40]. In addition, when chitosan reacts with multi-walled carbon nanotubes, it leads to the shifting of Raman spectra.

#### 3.1.3. FE-SEM-EDS Analysis

The morphological features and agglomeration of fabrication inks were determined by FE-SEM equipped with EDS. The samples of fabrication ink were subjected to JEOL JSM7800F field emission scanning electron microscopy at 2 kV for two compositions of Nano-ink with and without silver nanoparticles (AgNPs). Figure 3a,b shows randomly interconnected distributions of MWCNTs. The diameter of the MWCNTs-Cs is approximately 20–25 nm. The surface morphology of MWCNTs/SiC/AgNPs (1:1:0)-ink fabricated by using chitosan with medium molecular weights (Mw = 125–500 kDa, %DD = 90–95%) shows higher surface roughness and porosity than ink prepared with high molecular weight chitosan (Figure 4b).

Figure 4c reveals that the surface morphology of ink fabricated using chitosan of medium molecular weight was still intact even after adding AgNPs. However, for high molecular weight chitosan, after adding 25 mg of AgNPs (ratio of 1:1:0.5), Figure 4d shows a smooth and dense surface, which is different from the presence of 50 mg of AgNPs (percentage of 1:1:1), which has a porous and rough surface texture (Figure 4e).

Figure 5a shows the material composition of fabrication ink. Fabrication ink, after the addition of silver nanoparticles, was characterized by energy dispersive spectroscopy (EDS). There were peaks of different materials, including C, O, Al, Si, and Ag, with composition percentages of 72.58%, 6.47%, 2.63%, 0.55%, and 17.76%, respectively. High carbon elements were present due to multi-walled carbon and silicon carbide substrates, while oxide and Al were present due to drying and contamination. Figure 5b shows energy-dispersive X-ray spectroscopy (EDS) elemental mapping of AgNPs in fabrication ink. The results show clearly that AgNPs were uniformly distributed in fabrication ink.

### 3.2. Electrochemical Measurements

Electrochemical investigations of MWCNTs/SiC/chitosan and MWCNTs/SiC/ AgNPs/chitosan were performed at room temperature. A total of 5 mmol dm^−3^ of K_3_[Fe(CN)_6_]/K_4_[Fe(CN)_6_] solution is used as the redox probe to monitor the electron transfer efficiency of the electrode surface. Nano-ink was fabricated by MWCNTs/SiC and chitosan (medium: 125–500 kDa and high: 500–900 kDa). The performance of the three electrodes was characterized by cyclic voltammetry.

In an electrochemical investigation of MWCNTs/SiC/AgNPs-Chitosan, the performance of this Nano-ink depends on the molecular weight of chitosan and the ratio of AgNPs. Figure 6 curves (a) and (b) were used to compare the performance of the molecular weight of chitosan in enhancing electrochemical performance. The results show that medium molecular weight (125–500 kDa) performed better than high molecular weight (500–900 kDa).

Figure 6 curves (c) and (e) consist of (c) MWCNTs/SiC/AgNPs—50 mg of Cs (Med) (1:1:1-Cs (Med)) and (e) MWCNTs/SiC/AgNPs—50 mg of Cs (High) (1:1:1-Cs (High)) and have better electrochemical performance. The comparison between Nano-ink with and without AgNPs shows that if the proper molecular weight is 125–500 kDa (Med), then MWCNTs/SiC/AgNPs-Cs (Med) will be used in subsequent tests for working electrodes.

### 3.3. Differential Pulse Voltammetry Studies

The electrochemical behavior of paraoxon was analyzed by a fabrication ink that consisted of multi-walled carbon nanotubes, silicon carbide, silver nanoparticles, and chitosan as working electrodes on screen-printed carbon electrodes. The schematic illustration of the interaction between electrode modifiers on the electrode surface and the mechanism for paraoxon detection is shown in Figure 7. The sensor selectivity is proven in an experiment in which only paraoxon has been used as the calibration reference and produces a specific peak.

Figure 8A shows that the DPV reduction current peak of paraoxon decreased with increased paraoxon concentration. Figure 8B shows a linear plot between the reduction current response and paraoxon concentration, which increases linearly with increasing paraoxon concentrations in the range of 0.001–0.1 µg/mL. This gives linear detection ranges (Y = 18.053x + 4.2204), with a correlation coefficient of 0.9959, a detection limit of 0.0038 µg/mL, and a quantification limit of 0.0115. The limit of detection (LOD) and limit of quantification (LOQ) were calculated in the linear range of the calibration curve according to the equations: LOD = ((3.3)SD)/m, and LOQ = (10(SD))/m
where SD is the standard deviation, and m is the slope of the calibration curve [15].

A fabrication ink that consisted of MWCNTs/SiC/AgNPs as working electrodes on screen-printed carbon electrodes was used to analyze the electrochemical behavior of paraoxon, and it was found that this method better reflects the material’s properties for this application of paraoxon detection. Table 2 shows a comparison between the organophosphate detection performance of Nano-ink-modified electrodes and the previously reported detection method.

## 4. Conclusions

In this study, we develop and characterize Nano-ink by adding silver nanoparticles to increase the efficiency of the application. Nano-ink was fabricated with multi-walled carbon nanotubes, silicon carbide, and silver nanoparticles synthesized by the green method with corn starch and a medium molecular weight of chitosan. Electrochemical measurements have confirmed that adding silver nanoparticles can improve the efficiency of pesticide detection. Moreover, the process is simple and makes it easy to detect pesticide residuals. The research plan regarding the follow-up of this article is to investigate paraoxon in the agricultural field and compare it with conventional methods such as HPLC, UV-Vis spectrophotometry, etc.

## Figures and Tables

**Figure 1 micromachines-14-01613-f001:**
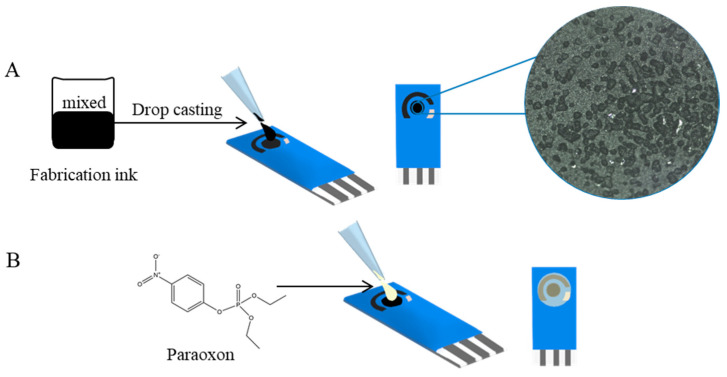
The preparation process for electrochemical analysis. (**A**) Preparing the working electrode and (**B**) pesticide solution to quantitatively detect paraoxon on the screen-printed carbon electrode.

**Figure 2 micromachines-14-01613-f002:**
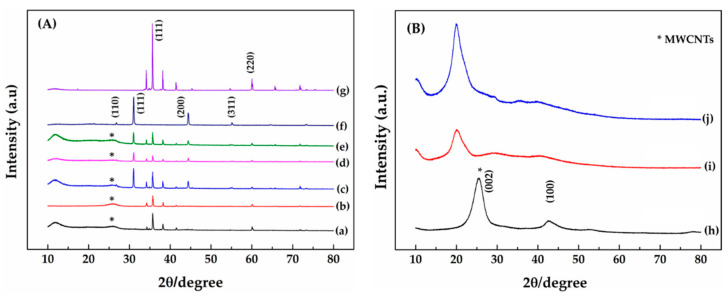
(**A**) XRD spectra of (a) 1:1:0—Cs (High), (b) 1:1:0—Cs (Med), (c) 1:1:1—Cs (High), (d) 1:1:1—Cs (Med), (e) 1:1:0.5—Cs (Med), (f) synthesis AgNPs, and (g) pure SiC. (**B**) XRD spectra of commercial (h) MWCNTs, (i) chitosan—Cs (Med), and (j) chitosan—Cs (High). Note that (*) represents the (002) plane of the MWCNTs’ position.

**Figure 3 micromachines-14-01613-f003:**
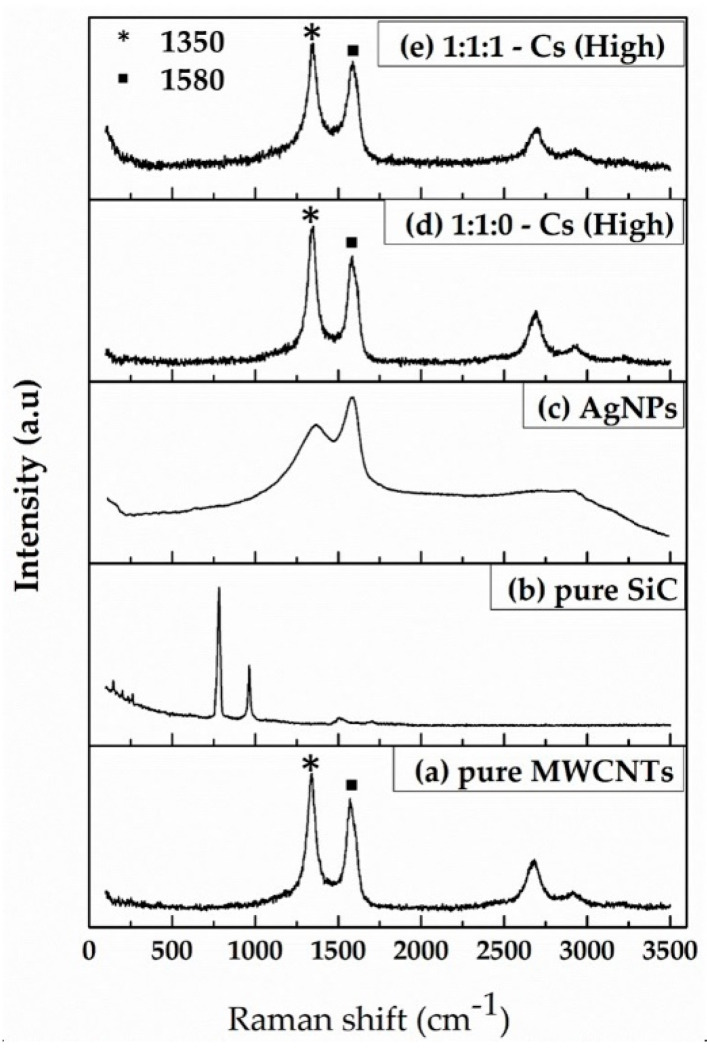
Raman spectra of (**a**) pure MWCNTs, (**b**) pure SiC, (**c**) pure AgNPs, (**d**) 1:1:0 of MWCNTs/SiC/AgNPs-Cs (High), and (**e**) 1:1:1 of MWCNTs/SiC/AgNPs-Cs (High).

**Figure 4 micromachines-14-01613-f004:**
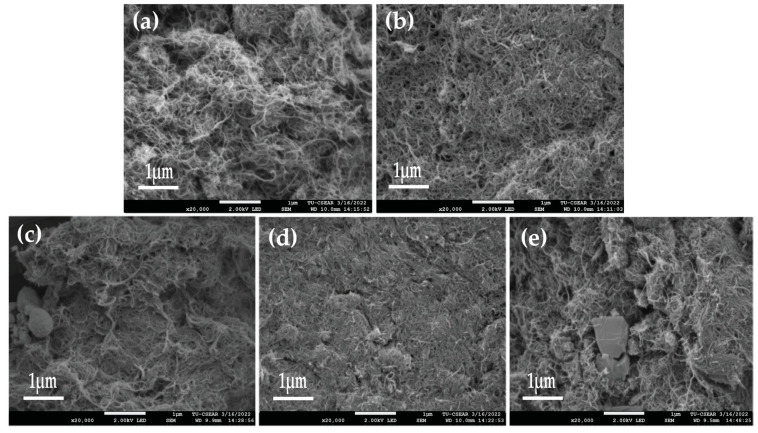
FESEM images of Nano-ink. (**a**) 1:1:0–Cs (Med), (**b**) 1:1:0–Cs (High), (**c**) 1:1:1–Cs (Med), (**d**) 1:1:0.5–Cs (High), and (**e**) 1:1:1–Cs (High).

**Figure 5 micromachines-14-01613-f005:**
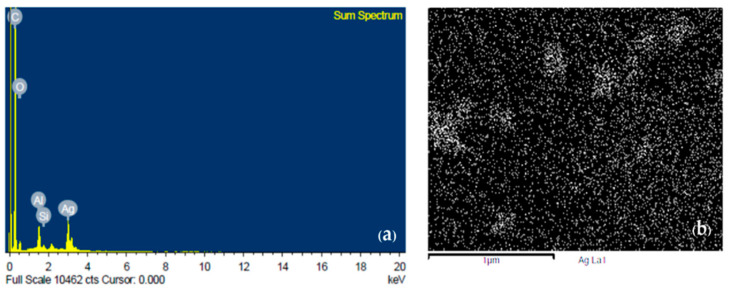
(**a**) Energy dispersive spectroscopy (EDS) shows visible peaks of C, O, Al, Si, and Ag; (**b**) silver nanoparticle dispersion mapping by energy-dispersive X-ray spectroscopy (EDS).

**Figure 6 micromachines-14-01613-f006:**
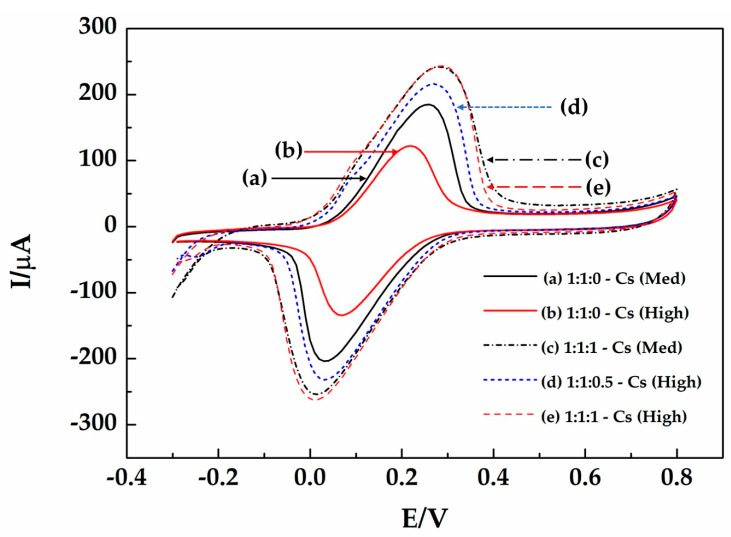
Cyclic voltammograms of the ink composed of MWCNTs/SiC/AgNPs as ratios of (a) 1:1:0-Cs (Med), (b) 1:1:0-Cs (High), (c) 1:1:1-Cs (Med), (d) 1:1:0.5-Cs (High), and (e) 1:1:1-Cs (High) in a 5 mmol dm^−3^ K_3_[Fe(CN)_6_]/K_4_[Fe(CN)_6_] solution containing 0.1 M KCl at a scan rate of 10 mVs^−1^.

**Figure 7 micromachines-14-01613-f007:**
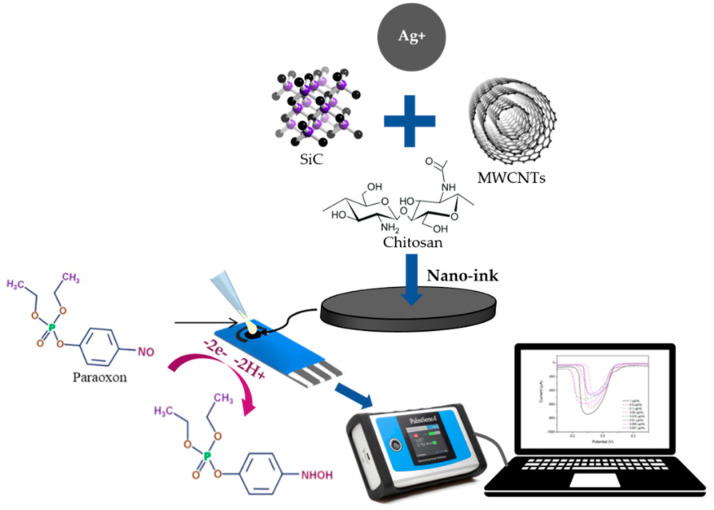
A schematic illustration of the interaction between electrode modifiers on the electrode surface and the mechanism for paraoxon detection.

**Figure 8 micromachines-14-01613-f008:**
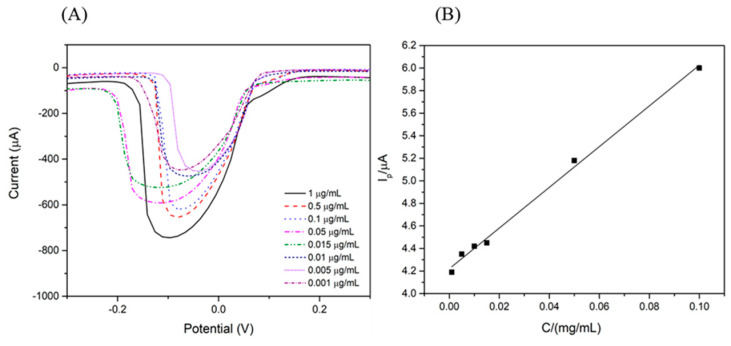
(**A**) Differential pulse voltammetry (DPV) response of paraoxon (0.001–1 µg/mL) in 0.1 M PBS of pH 7.0 at a scan rate of 25 mVs^−1^. (**B**) Calibration plot of the current response vs. the concentration of paraoxon (0.001–0.1 µg/mL) in 0.1 M PBS of pH 7.0 at a scan rate of 25 mVs^−1^.

**Table 1 micromachines-14-01613-t001:** The composition ratio of MWCNTs/SiC/AgNPs and the molecular weight of chitosan.

Sample	The Ratio of MWCNTs/SiC/AgNPs	Mw of Chitosan (Medium/High: mg)
1	1:1:0	Medium: 50
2	1:1:0	High: 50
3	1:1:0.5	High: 50
4	1:1:1	Medium: 50
5	1:1:1	High: 50

**Table 2 micromachines-14-01613-t002:** Comparison between the organophosphate detection performance of Nano-ink-modified electrodes and the previously reported detection method.

Nanomaterials	OP Pesticide	Linear Concentration Range	LOD	Sensitivity	Detection Technique	Electrode Type	Ref.
ZrO_2_ NPs	Paraoxon	5–100 ng/mL	3 ng/mL	-	SWV	GCE	[41]
MWCNTs	Parathion	0.05–2.0 μg/mL	0.005 μg/mL	-	SWV	GCE	[42,43]
SiCNPs/MWCNTs/ Chitosan	Parathion	0–10,000 ng/mL	20 ng/mL	0.00198 and 0.0006975 μAL/mg	DPV	-	[15]
Graphene-NiFeSP	Paraoxon	0.01–1.00 M and 1.00–10.00 M	3.7 nmol/L	10.243 and 2.667 μAL/μmol	SWV	GCE	[44]
BiVO4	Paraoxon	0.2–1.96 M	0.034 μM	0.345 μA /μM∙cm^2^	DPV	Print screen carbon	[45]
GONS@CuFeS2	Paraoxon	0.073–801.5 μM	4.5 nM	17.97 μA/μM∙cm^2^	DPV	Print screen carbon	[45]
Nano-ink (MWCNTs/SiC/AgNPs)	Paraoxon	0.001–0.1 μg/mL	0.0038 μg/mL	18.053 μAL/g	DVP	Print screen carbon	This work

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
