# Peer review of "Development and Characterization of Nano-Ink from Silicon Carbide/Multi-Walled Carbon Nanotubes/Synthesized Silver Nanoparticles for Non-Enzymatic Paraoxon Residuals Detection"

_micromachines, 2023, doi:10.3390/mi14081613_

Round 1

Reviewer 1 Report

(1)Line 91, 5-FU should clarify the meaning of the acronym. And the fourth paragraph of the introduction section about Silver nanoparticles(Ag NPs) does not read clearly and smoothly, and the expression needs to be revised.

(2)The figure legend of Figure 2 does not indicate the meaning of c.

(3)In line 240, the manuscript mentions that electrode (a) will be compared to Silver nanoparticles added in the nano-link. Still, I do not see a comparison between these two electrodes regarding electrochemical performance.

(4)In line 262, the manuscript mentions that the linearity range is 0.001μg/mL-1μg/mL, but in Figure 7(b), the concentration of paraoxon only goes up to 0.1μg/mL. And in the next line, the article mentions that the detection limit is 0.0038, but in the abstract, it says that the limit of detection is 0.0037. You need to modify it to ensure consistency.

(5)In the conclusion section, I would like you to describe the research plan regarding the follow-up of this article.

In line 276, I don't know what the word detectusedesticide means

Author Response

Reviewer #1

Comment

Answer

(1) Line 91, 5-FU should clarify the meaning of the acronym. And the fourth paragraph of the introduction section about Silver

Nanoparticles (Ag NPs) does not read clearly and smoothly, and the expression needs to be revised.

This has been revised accordingly.

Line 109

(2) The figure legend of Figure 2 does not indicate the meaning of c.

This has been revised accordingly.

Line 236-237

(3) In line 240, the manuscript mentions that electrode (a) will be compared to silver nanoparticles added in the nano-link. Still, I

do not see a comparison between these two electrodes regarding electrochemical performance.

This has been revised accordingly.

Line 277-291

(4) In line 262, the manuscript mentions that the linearity range is 0.001μg/mL-1μg/mL, but in Figure 7(b), the concentration of

paraoxon only goes up to 0.1μg/mL. And in the next line, the article mentions that the detection limit is 0.0038, but in the abstract, it says that the limit of detection is 0.0037. You need to modify it to ensure consistency

This has been revised accordingly.

Line 301-311

We added information and equation according to reviewer suggestion:

LOD = ((3.3)SD)/m,

and LOQ = (10(SD))/m

(5) In the conclusion section, I would like you to describe the research plan regarding the follow-up of this article.

This has been revised accordingly.

Line 324-326

To investigate paraoxon in the agricultural field and do compare it with the conventional method such as HPLC, GC-MS, etc.

(6) In line 276, I don't know what the word detectusedesticide means

This has been revised accordingly.

Line 323-324

Reviewer 2 Report

-The performance parameters of this sensors should be compared to those of other sensor for paraoxon to better illustrate the novelty and usefulness of the sensor material.

-Lines 263 and 264, the units of measurement for the limits of detection should be specified. 

-Two linear ranges are mentioned but only one equation and one calibration curve are mentioned. Also the ranges themselves are not specified.

-A comparison of the response for paraoxon between the various sensors tested should be provided. In its current form the work does not provide any sufficient proof the chosen material is indeed suitable for paraoxon detection. CV or DPV in paraoxon (not just in ferry-ferrocyanide) could be shown

Author Response

Reviewer #2

Comment

Answer

(1)   The performance parameters of this sensors should be compared to those of other sensor for paraoxon to better illustrate the novelty and usefulness of the sensor material.

This has been mentioned in the conclusion session.

Line 332-335

To investigate paraoxon in the agricultural field and do compare it with the conventional method such as HPLC, UV-Vis spectrophotometry, etc.

(2)   Lines 263 and 264, the units of measurement for the limits of detection should be specified.

This has been revised accordingly.

Line 310-320

(3)   Two linear ranges are mentioned but only one equation and one calibration curve are mentioned. Also, the ranges themselves are not specified.

This has been revised accordingly.

Line 310-320

We added information and equation according to reviewer suggestion:

LOD = ((3.3)SD)/m,

and LOQ = (10(SD))/m

(4)  A comparison of the response for paraoxon between the various

sensors tested should be provided. In its current form the work does not provide any sufficient proof the chosen material is indeed suitable for paraoxon detection. CV or DPV in paraoxon (not just in ferry-ferrocyanide) could be shown

This has been mentioned in the conclusion session.

Line 332-335

To investigate paraoxon in the agricultural field and do compare it with the conventional method such as HPLC, UV-Vis spectrophotometry, etc.

Reviewer 3 Report

In this manuscript by Chuasontia et al. described the development of a nonenzymatic sensor based on a composite of metal nanoparticles-silicon carbide-carbon nanomaterials for paraoxon detection. Several instrumentation techniques were employed to characterize this nonenzymatic sensor. The results are interesting. However, there are many important issues that need to be revised to enrich the manuscript quality as well as to give a better understanding of the work as follows:

1.          Why did the author choose the word “nano-ink” in the title of the manuscript? Is there any justification for the particle size of this composite characterized by the instrumentation technique such as particle size analyzer (PSA)?

2.          In line 68, the sentence “silicon carbide-modified electrode….” needs to cite the published paper to emphasize the importance of using this material as an electrode modifier. The previous work about electrodes based on silicon carbide needs to be added more in the introduction section as it gives strong background on why the author choose this material to be developed further as an electrochemical sensor.

3.          In paragraph 3 in the introduction section, why does the author explain a lot about chitosan? What is the most important reason for employing this material for the synthesis of silver nanoparticles?

4.          In the materials and method section, the type of potentiostat employed in this work needs to be addressed appropriately.

5.          In materials and method, the type of Raman spectrometer used in this work needs to be written in detail.

6.          In the materials and method section, a commercial screen-printed carbon electrode does not contain silicon nanoparticles and MWCNT ink as the working electrode, Ag/AgCl saturated with KCl as a reference electrode. The authors must be very careful in writing the experimental conditions, especially in the method and materials section.

7.          In Figure 1A, there is a zoom image in the working electrode, is this coming from a light microscope or SEM? Is there any magnification from this image? It looks like a lot of small particles or granules on the surface. Where do these particles come from?

8.          In the results and discussion section, why did the author choose only 4 ink samples to be analyzed with XRD? While in Table 1, the authors prepared 7 different ratios of inks to be investigated.

9.          Is there any diffractogram (XRD spectra) for pure MWCNT, SiC, AgNPs, and chitosan with high or medium molecular weight?

10.       In Raman studies (Figure 3), if 2 Raman peaks for pure SiC (≈ 750 and 1000 cm-1) disappear when SiC was combined MWCNT-AgNPs and chitosan, maybe the y-axis for intensity in Figure 3 needs to be rescaled.

11.       In Raman studies (Figure 3), which ratio of MWCNT-SiC-AgNPs-Cs was analyzed and investigated with Raman spectroscopy?

12.       Line 211, Raman spectrum of AgNPs does not contain any peak. Why is it written 2 Raman peaks for AgNPs? Is this coming from a composite of AgNPs-chitosan?

13.       In Figure 4, what is the significant difference between Figures 4A, B, and C?

14.       it is also hard to see any formation of AgNPs in Figures 4D, E, and F unless it can be confirmed with the data from the EDS mapping element.

15.       In Figure 5, is there any data from the EDS mapping element for the SEM image of MWCNT-SiC-AgNPs to confirm the distribution of AgNPs on the ink surface?

16.       In electrochemical measurements, is there any CV data for the measurement of K3[Fe(CN)6] using a bare screen-printed carbon electrode (SPCE)? The authors need to show this data and compare it with the modified SPCE as it is expected the usage of material modifiers (MWCNT, SiC, AgNPs, Chitosan) to enhance the electrode sensitivity. Are there any electrochemical impedance spectroscopy (EIS) studies to support the CV findings?

17.       Are there any fundamental parameters that have been characterized and investigated from these modified electrodes? The fundamental parameters can be characterized such as the effective surface area (ECSA) of each electrode, the effect of scan rate, the electron transfer coefficient, the number of transferred electrons, the surface coverage of electrode, and the diffusion coefficient.

18.       Several analytical performances of the modified electrode can be investigated using the modified electrode to enrich the manuscript quality and to better understand this electrode's employability. The authors are advised to determine the limit of detection (LOD), the limit of quantification (LOQ), and the sensitivity of each electrode for paraoxon detection.

19.       It is suggested to do more experiments to determine the selectivity of this sensor and the detection of paraoxon in real samples using the DPV technique. The result of paraoxon detection using these modified electrodes can be further compared with other standardized techniques such as UV-Vis spectrophotometry or HPLC.

20.       It is also suggested to write the proposed mechanism for paraoxon detection using these modified electrodes.

21.       Finally, drawing the schematic illustration of the interaction between all electrode modifiers on the electrode surface with paraoxon will help the readers understand them quickly.  

There are many grammatical mistakes in this manuscript and it is strongly recommended to polish the manuscript.

Author Response

Reviewer #3

Comment

Answer

(1) Why did the author choose the word “nano-ink” in the title of the manuscript? Is there any justification for the particle size of this composite characterized by the

instrumentation technique such as particle size analyzer (PSA)?

The nano-ink was referred from the nano size of the surface coating according to the FESEM image of Nano-ink. The size of the coating particle is lower than 1 micrometer, as shown in the scale bar of FESEM image.

(2)   In line 68, the sentence “silicon carbide-modified electrode….” needs to cite the published paper to emphasize the importance of using this material as an

electrode modifier. The previous work about electrodes based on silicon carbide needs to be added more in the introduction section as it gives strong background on why the author choose this material to be developed further as an electrochemical sensor.

This has been revised accordingly.

Line 85

(3) In paragraph 3 in the introduction section, why does the author explain a lot about chitosan? What is the most important reason for employing this material for the

synthesis of silver nanoparticles?

This has been revised accordingly.

Line 81-118

We add more details about Silicon carbide (SiC) and multiwall carbon nanotubes. Moreover, it has been described the reason why chitosan is so important.

(4) In the materials and method section, the type of potentiostat employed in this work needs to be addressed appropriately

This has been revised accordingly.

Line 151-154

(5) In materials and method, the type of Raman spectrometer used in this work needs to be written in detail.

This has been revised accordingly.

Line 147-149

(6) In the materials and method section, a

commercial screen-printed carbon electrode does not contain silicon nanoparticles and MWCNT ink as the working electrode, Ag/AgCl saturated with KCl as a reference electrode. The authors must be very careful in writing the experimental conditions, especially in the method and materials section.

This has been revised accordingly.

Line 179-184

(7) In Figure 1A, there is a zoom image in the

working electrode, is this coming from a light microscope or SEM? Is there any magnification from this image? It looks like a lot of small particles or granules on the

surface. Where do these particles come from?

This has been revised accordingly.

Line 188-196

(8) In the results and discussion section, why did the author choose only 4 ink samples to be analyzed with XRD? While in Table 1, the authors prepared 7 different ratios of inks to be investigated.

This has been revised accordingly in

3.1.1. X-ray diffraction (XRD) and change Figure2 to Figure2A and 2B.

Line 203-217 and line 235-239

(9) Is there any diffractogram (XRD spectra) for pure MWCNT, SiC, AgNPs, and chitosan with high or medium molecular weight?

This has been revised accordingly in

3.1.1. X-ray diffraction (XRD) and change Figure2 to Figure2A and 2B.

Line 203-217 and line 235-239

(10) In Raman studies (Figure 3), if 2 Raman peaks for pure SiC (≈ 750 and 1000 cm-1) disappear when SiC was combined MWCNT-AgNPs and chitosan, maybe the yaxis for intensity in Figure 3 needs to be rescaled.

This has been revised accordingly.

3.1.2. Raman Spectroscopy   

Line 218-233 and 241-242

(11) In Raman studies (Figure 3), which ratio of MWCNT-SiC-AgNPs-Cs was analyzed and investigated with Raman spectroscopy?

This has been revised accordingly.

3.1.2. Raman Spectroscopy   

Line 218-233 and 241-242

(12) Line 211, Raman spectrum of AgNPs does not contain any peak. Why is it written 2 Raman peaks for AgNPs? Is this coming from a composite of AgNPs-chitosan?

This has been revised accordingly.

Line 244-245

(13) In Figure 4, what is the significant difference between Figures 4A, B, and C?

This has been revised accordingly.

Line 252-262

(14) It is also hard to see any formation of AgNPs in Figures 4D, E, and F unless it can be confirmed with the data from the EDS mapping element.

This has been revised accordingly.

Line 263-270

(15) In Figure 5, is there any data from the EDS mapping element for the SEM image of MWCNT-SiCAgNPs to confirm the distribution of AgNPs on the ink surface?

This has been revised accordingly.

Line 263-270

(16) In electrochemical measurements, is there any CV data for the measurement of K [Fe(CN) ] using a bare screen-printed carbon electrode (SPCE)? The authors need to show this data and compare it with the modified

SPCE as it is expected the usage of material modifiers (MWCNT, SiC, AgNPs, Chitosan) to enhance the electrode sensitivity. Are there any electrochemical impedance spectroscopy (EIS) studies to support the CV

findings?

This has been revised accordingly.

Line 278-297

(17) Are there any fundamental parameters that have been characterized and investigated from these modified electrodes? The fundamental parameters can be characterized such as the effective surface area (ECSA) of each electrode, the effect of scan rate, the electron transfer coefficient, the number of transferred electrons, the surface coverage of electrode, and the diffusion coefficient.

We plan to do more characterizing and investigating in the future.

(18) Several analytical performances of the modified electrode can be investigated using the modified electrode to enrich the manuscript quality and to better understand

this electrode's employability. The authors are advised to determine the limit of detection (LOD), the limit of quantification (LOQ), and the sensitivity of each electrode

for paraoxon detection.

This has been revised accordingly.

Line 310-320

(19) It is suggested to do more experiments to determine the selectivity of this sensor and the detection of paraoxon in real samples using the DPV technique. The result of paraoxon detection using these modified

electrodes can be further compared with other standardized techniques such as UV-Vis spectrophotometry or HPLC.

This has been mentioned in the conclusion session.

Line 332-335

To investigate paraoxon in the agricultural field and do compare it with the conventional method such as HPLC, UV-Vis spectrophotometry, etc.

(20) It is also suggested to write the proposed

mechanism for paraoxon detection using these modified electrodes.

This has been revised accordingly.

Line 301-307

(21) Finally, drawing the schematic illustration of the interaction between all electrode modifiers on the electrode surface with paraoxon will help the readers understand them quickly

This has been revised accordingly.

Line 301-307

(22) There are many grammatical mistakes in this manuscript and it is strongly recommended to polish the manuscript.

This has been revised accordingly.

Round 2

Reviewer 2 Report

Even though improvements were made to the manuscript, the important issue of proving the quality of the sensor material for detecting paraoxon specifically still remains. Section 3.2 only shows a general electrocatalytic effect of the material, and it could be assumed that this would achieve the detection of a wide variety of compounds, not just paraoxon, so the sensor selectivity is not sufficiently proven. A comparison of CVs and/or DPVs in paraxon with sensors modified with several materials (for example MWCNT, MWCNTs/AgNPs, SiC/AgNPs …) could be compared to those of the MWCNTs/SiC/AgNPs to better show the properties of this material for this application.

It is better to add this data to the article, otherwise it would not be appropriate to conclude that the MWCNTs/SiC/AgNPs material is indeed useful for paraoxon detection, and/or a comparison of the detection limit and the slope (sensitivity) achieved with other sensors in the literature should be provided.

The statement in the Conclusion:

“Electrochemical measurements have confirmed that adding silver nanoparticles can improve and enhance the efficiency of pesticide detection.” is not justified. At most it could be concluded that the sensors with this material can detect paraoxon but not how effective they are compared to sensors with other materials.

Author Response

(1) Section 3.2 only shows a general electrocatalytic effect of the material, and it

could be assumed that this would achieve the detection of a wide variety of compounds, not just paraoxon, so the sensor selectivity is not sufficiently proven.

This has been revised accordingly.

Line 308-310

(2) A comparison of CVs and/or DPVs in paraxon with sensors modified with several materials (for example MWCNT, MWCNTs/AgNPs, SiC/AgNPs …) could

be compared to those of the MWCNTs/SiC/AgNPs to better show the properties of this material for this application.

This has been revised accordingly.

Line 332-340 and Table 2.

Reviewer 3 Report

Accept

Author Response

-

Round 3

Reviewer 2 Report

Even though a comparison of the CVs for paraoxon for sensors with several materials should have been provided, useful changes were made to the manuscript, such as adding the comparison with the literature, and the article can be accepted for publishing.